# The Formability of Perforated TA1 Sheet in Single Point Incremental Forming

**DOI:** 10.3390/ma16083176

**Published:** 2023-04-18

**Authors:** Ruxiong Li, Tao Wang, Feng Li

**Affiliations:** 1Faculty of Materials Science and Technology, Nanjing University of Aeronautics and Astronautics, Nanjing 211100, China; 2Faculty of Mechatronics Engineering, Jingdezhen Ceramic University, Jingdezhen 333403, China

**Keywords:** perforated TA1 sheet, single point incremental forming, wall angle, fracture mechanism, numerical simulation

## Abstract

In light of the analysis on the single point incremental forming (SPIF) principle of perforated titanium sheet and the corresponding peculiarities during the forming process, it is found that the wall angle constitutes the pivotal parameter influencing the SPIF quality of the perforated titanium sheet, and this is also the key evaluation index to test the application of SPIF technology on a complex surface. This method for integrating the experiment and the finite element modelling was utilized in this paper to study the wall angle range and fracture mechanism of Grade 1 commercially-pure α titanium (TA1) perforated plate, plus the effect of different wall angles on the quality of perforated titanium sheet components. The forming limiting angle, fracture, and deformation mechanism of the perforated TA1 sheet in the incremental forming were obtained. In accordance with the results, the forming limit is related to the forming wall angle. When the limiting angle of the perforated TA1 sheet in the incremental forming is around 60 degrees, the fracture mode is the ductile fracture. Parts with a changing wall angle have a larger wall angle than parts with a constant angle. The thickness of the perforated plate formed part does not fully satisfy the sine law, and the thickness of the thinnest point of the perforated titanium mesh with different wall angles is lower than that predicted by the sine law; therefore, the actual forming limit angle of the perforated titanium sheet should be less than that predicted by a theoretical calculation. With the increase in the forming wall angle, the effective strain, the thinning rate, and the forming force of the perforated TA1 titanium sheet all increase, while the geometric error decreases. When the wall angle of the perforated TA1 titanium sheet is 45 degrees, the parts with a uniform thickness distribution and good geometric accuracy can be obtained.

## 1. Introduction

Titanium alloy is usually a low plastic material, so it is difficult to process it through plastic deformation. SPIF technology is a highly flexible computer numerical control (CNC) sheet metal forming technology that can quickly form thin-walled workpieces of complicated shapes without the help of special mold. It has attracted wide interest in the research community because of its enhanced formability, flexible manufacturing ability, and reduced forming force. Thus, SPIF has greater potential in the aspects of rapid prototype manufacturing as well as small batch production [1,2]. SPIF can fulfil the current requirements for flexible, feasible, sustainable, and economic manufacturing technology for personalized sheet metal parts, small-batch prototypes, and customized products without the use of expensive specialized machines or equipment [3,4,5].

In the SPIF process, the tool squeezed the titanium plate layer-by-layer along the forming contour line (*m*-direction) to finally form the truncation cone. The vertical distance (*n*-direction) between each adjacent contour line can be defined as the step-down size *s*, which is the angle between the deformation plate, and the initial plate can be defined as the wall angle *θ*. The tool completes the forming motion of contour advance within the layer and vertical downward pressure between layers along the contour trajectory, and the forming principle is shown in Figure 1. According to the deformation characteristics of the material, the main deformation area of the plate is divided into bending transition zone (*AB* area), formed area (*BC* area) and contact deformation area (*CD* area).

Hagan and Jeswiet [6] compared forming methods of sheet metal to illustrate the different characteristics of incremental forming processes. The local contact between the tool and the workpiece makes the forming force smaller, but the limit deformation of SPIF is greater than stamping. Based on the proposed technical features, SPIF is ideal for the customization of small batches or forming parts with complex geometries, and occupies an important position in rapid prototyping [7,8,9,10]. Jeswiet et al. [11] used the SPIF method to produce a prototypes model in the automotive industry. In medical applications, Verbert et al. [12] use multi-step incremental molding to manufacture titanium skull implants. Iseki and Naganawa [13] introduced the use of multi-level increments to form vertical thin wall surfaces. Jeswiet et al. [14] describe improvements to traditional sheet metal forming, with an emphasis on asymmetric single-point incremental forming, providing valuable guidance to designers and manufacturers. Ambrogio et al. [15] successfully introduced the incremental forming to produce a customised ankle support with good dimensional accuracy, which has shown good results for the desired application. Behera et al. [16] clarified the shape and accuracy of titanium medical implants in the incremental forming process. Hagan and Jeswiet et al. [6] conducted a literature review of the SPIF process that covers the findings obtained before 2005, and Behera et al. [17] combined the findings reported from 2005 to 2015.

Feasibility studies of making biomedical implants by SPIF techniques have become more common. In this context, citing work [18,19,20,21] considering the shortcomings of the current manufacturing process, the final state of prosthesis demonstrates that the forming quality of restorations formed by the SPIF can meet the needs of patients and replace missing human body parts, which may occur due to degenerative diseases, traumatic accidents, or tumors.

The formability of sheet metal forming, which is one of the very important responses in the SPIF process, is the ability of the metal to deform without exhibiting a specific form of failure. To correctly and safely form the components, the maximum formability of the component in the SPIF process is usually estimated. The metal plate is gradually formed, and the formability and potential of materials can thus be fully explored [14,22].

The forming limit diagram (FLD), which was introduced by Keeler and Backofen [23] and given an analytic basis by Marcinak and Kuczynski [24], have been used to study and describe the maximum strain under various strain paths in sheet deformation, and it can represent the formability in SPIF. The forming limit curve (FLC) is usually employed to determine the limit of proportional strains before the sheet fracturing under a variety of strain ratios, but the corresponding one in traditional forming cannot represent the formability in all straining conditions. Indeed, it is known that the FLC is only valid if it satisfies four conditions: the straight strain path (proportional loading), the deformation induced by membrane forces without bending, ignoring the through-thickness shear, and the plane stress perpendicular to the sheet surface [25]. SPIF is a process of stretching combined with shear forces, normal or bending forces, and cyclic or non-proportional deformation paths, and the suggested explanations, indeed, violate the four conditions, which conflicts with the four conditions of the standard FLC.

Conventionally, a series of frustums of the cone or pyramids are used for formability measurements, finally achieving the maximum wall angle of the forming parts. The wall angle is generally decreased in small steps until the sheet endures a maximum angle without fracture [26]. Young and Jeswiet [27] proposed a double-pass technique to test the wall thickness profiles of a series of cones, and whilst the test results show that the wall thickness profiles were unstable, a significant thinning band appeared when approaching the maximum wall angle, which was also confirmed by Salem [28]. Hussain and Gao [10,29] did similar work on the different corners of a cone. They found that the maximum wall angle of cones obtained in the funnel with fixed slopes was a litter lower than those cones with fixed wall angle. Therefore, the above studies showed that when setting for the incremental forming, it is recommended that the maximum wall angle of a given material is used as an indicator of the formability of the material.

Over the past two decades, extensive research on SPIF has made significant progress in the basic understanding and development of new processing and processing solutions. However, SPIF has not yet been fully implemented in the mainstream of the high-value manufacturing industries due to some technical challenges. These challenges are all directly related to the process parameters associated with the SPIF.

The importance of the four variables, i.e., sheet thickness, tool diameter (dp), wall angle (θ), and step-down size (s), are well known [30,31]. Gate et al. [32] reviewed the technical capabilities and specific limitations of the current incremental forming process. The process parameters and their impact on the incremental forming process are discussed. Particular attention is paid to the effects of incremental process parameters on formability, deformation and failure mechanics, rebound, precision, and surface roughness. Eeyckens et al. [33] suggested that the main deformation mechanism of the incremental formation is correlated to the chosen forming parameters, where the most basic method to determine the incremental formation is to use the wall angle of the part to be formed as the limiting factor. Silva et al. [34] used SPIF to analysis the thickness distribution of the formed sheet. The measurement of thickness and true strain showed that the titanium sheet with 0.5 mm thickness has the limit wall angle of 47 degrees. The thickness of the workpiece obtained by the sine law should be 0.35 mm, but the experimental result was approximately 0.25 mm.

Based on experimental work and finite element (FE) simulation, scholars have proposed a number of mechanisms to explain the enhanced formability in SPIF [35,36]. It has been suggested that the deformation mechanism for SPIF is stretch and shear in a plane that is perpendicular to the direction of tool movement, together with shear in a plane that is parallel to the direction of the tool movement. It is also found that the measured shear strain in the direction of the tool motion was the greatest strain component.

Kumar et al. [37] performed experimental work on the formability of AA2024-O aluminum alloy sheets by SPIF process. They delved into the mechanism of forming forces and came to conclusions that the sheet thickness was the third most dominating factor affecting the formability of the SPIF whilst the wall angle and the step-down size have been proved to be more significant factors affecting the formability of the SPIF. Eeyckens et al. [33] used both the experimental basis and the numerical basis method to perform an in-depth analysis of the strain distribution in the SPIF process. The results showed that the dominant deformation mechanism of SPIF mainly depends on the wall angle and step-down size, where the most basic method to determine formability in SPIF is to use the wall angle of the part to be formed as a limiting factor. Due to a limited quantity of materials available, the step-down size and sheet thickness were removed from the study to reduce the number of tests.

The forming force has a great influence on the fracture mechanism and accuracy of the forming part. Under the action of the forming force, stress and strain are created in the sheet. Nevertheless, stress is not only related to the forming force but also the plastic strain, which further calculates the structural integrity of the final component [38]. To ensure the safety of the forming equipment, it is necessary to study the maximum forming force of SPIF process. Furthermore, a knowledge of the magnitude of the forming force at the point where the tool and the sheet come into contact would greatly help to improve the forming performance of the SPIF process.

Duflou et al. [39] predicted the forming forces for the parts with complex geometries. On the basis of the experiments, the relationship between the forces for parts with uniform wall angles and the main process parameters were established. A regression equation of the forces was obtained in terms of the step-down size, tool diameter, wall angle, and sheet thickness, which can predict for the peak, steady-state, and in-plane forces.

Liu et al. [40] compared the effects of different draw angles (60 degree, 65 degree and 70 degree) and the type of tool paths on AA7075-O aluminum alloy sheets. The calculated results showed that different draw angles can affect their formability. The truncation cone with a draw angle of 60 degrees can be successfully formed. The trend of the resultant peak force in the initial phase was to increase with the draw angle, while the slope of the force curve after peak value showed the opposite trend. Li et al. [41] proposed an efficient analytical model for tangential force prediction and verified it experimentally on the sheets made by AA7075-O. The results showed that when the wall angle is greater than 60 degrees, the values of forming force peaks at a depth near 12 mm, and then moves monotonically towards the process failure. Chang et al. [42] proposed a new analytical model for predicting the forming force for SPIF, multi-pass incremental forming (MPIF), and incremental hole flanging (IHF) processes, which can be obtained by multiplying the contact area with the through-thickness stress. This proves that the proposed analytical models have wider applicability.

Arfa et al. [43] and Henrard et al. [44] used a finite element analysis to predict the forces in the SPIF process with satisfactory accuracy. The experimental and simulation results showed that the wall angle caused changes in material stress and strain conditions. The change of wall angle resulted in changes in material stress and strain conditions.

However, the above studies only focus on conventional plates rather than perforated titanium sheets with special structures, and although the formability of the material can be quantified by the limit wall angle, the definition of this formability does not include an understanding of the formation mechanism in its interpretation.

Compared with titanium plates, the perforated structure is more favorable for the growth of granulation tissue, and the internal interconnected and multidirectional pores produce capillary penetration, which is more conducive to the absorption of subcutaneous effusion [45]. Mechanistic studies of SPIF focus only on traditional plates rather than mesh plates, and although the formability of the material can be quantified using the largest wall angle, this definition of formability does not include an understanding explanation of the formation mechanism. The analysis and reach on SPIF for the perforated titanium sheet objects and the SPIF mechanism related to the perforated titanium sheet are rarely reported. Experimental research is the basic method to explore the SPIF law. However, there are great differences between the titanium plate and perforated titanium sheet in terms of physical properties and mechanical properties. Furthermore, SPIF is achieved by the local deformation of continuous changes. Therefore, based on the experimental research and simplified theoretical analysis, it is particularly difficult to explain the SPIF mechanism of the perforated titanium sheet from the mechanical aspects.

The study presented in this paper complements the previous research that started in 2017. The current research focus is on addressing the difficulties encountered in the SPIF process of forming biomedical implants with the perforated titanium sheet. The literature [46] constructed the truncated cone finite element model of the TA1 titanium plate and the perforated TA1 titanium sheet. Combined with theoretical analysis, the stress and strain changes and distribution rules of different regions during the SPIF process of titanium plate and titanium mesh were studied, and the mechanism of single point incremental forming of titanium mesh was revealed. At the same time, based on the established model, the distribution laws of the displacement field, wall thickness, and geometric errors in the SPIF process of the perforated titanium sheet were analyzed. The perforated TA1 sheet can be completed in the hole domain, so the out-of-plane deformation is much less than that of the titanium plate. The perforated titanium sheet structure can improve the accuracy of the member contour more effectively than the titanium plate. However, the literature [46] only analyzed the metal deformation behavior of the perforated titanium sheet. The formability and fracture mechanism of perforated titanium sheets have not been studied, and the influence of the forming wall angle on the forming quality has also not been analyzed.

This paper pays special attention to the formability, deformability, and accuracy of TA1 perforated titanium sheet. Experiment and finite element simulation (FEM) are used to determine the wall angle range of a perforated TA1 titanium sheet, and the fracture mechanism is analyzed by the fracture morphology. The influence of different wall angles on the stress strain and the wall thickness distribution within the overall volume of the SPIF process of the perforated titanium sheet are studied, and the forming force and energy in the main deformation area are discussed to obtain the SPIF mechanism of the perforated titanium plate. The results provide technical guidance for the shape design of the perforated titanium sheet based on the wall angle. Moreover, the results will also provide basic theoretical support for the analysis of the SPIF process of the perforated titanium sheet and provide important instructions useful for future work and the production of perforated titanium sheet biomedical implants with the SPIF process.

## 2. Materials and Methods

### 2.1. Material

The biomedical pure α titanium (TA1) (which are certificated for medical applications according to the DIN ISO 5832-2 norm) has been utilized in this study. The chemical composition is shown in Table 1.

### 2.2. Design of Experiments

In another study by Li and Wang [46], it was found that the TA1 sheets generally have anisotropic characteristics. To consider the effect of the anisotropic material model on the SPIF process, this paper refers to the literature [47,48,49], and the elastoplastic constitutive equation that meets the Hill 48 (Hill anisotropy yield criterion) can characterize the deformation characteristics of the titanium plate of cold-rolled steel, which is because cold rolling is often assumed to be homogeneous along the same plane of the plate but is anisotropic in the plate thickness. The Hill anisotropy yield criterion was used in the study of the SPIF process. Therefore, in order to establish a reasonable constitutive model of TA1 titanium alloy, the TA1 plate was cut by water jet cutting at an angle of 0 degrees, 45 degrees, and 90 degrees from the rolling direction of the plate to obtain three groups of tensile samples in different directions, and its mean values were taken as the material attribute parameter of the TA1 plate material [50]. The dimensions of the normalized specimens are depicted in Figure 2b. The thickness of the TA1 plate is 1 mm.

According to the standard GB/T 228-2010, the corresponding dimensions were measured using a contour projector to obtain the normal anisotropy coefficient for each rolling direction, as presented in Figure 2. By performing tensile tests on the titanium alloy material, the strain hardening index, strain hardening coefficient, elastic modulus, and thickness anisotropy coefficient can be used to define the elastoplastic behavior of materials in FEM simulations.

As shown in Figure 3, the SANS universal testing machine (SSANS Inc., Shanghai, China) with a loading tensile rate of 2 mm/min is adopted in the tensile test, and TA1 flow behavior was characterized on the base of the stress-strain curves that were obtained at room temperature after data processing. The TA1 plate rapidly enters the elastic deformation stage under the tensile force. After reaching the yield strength, the TA1 plate passed through a nearly uniform plastic deformation stage, in which the tensile sample was extended by about 17 cm. In this stage, the strain gradually increases, and the work hardening enhances the ability of the TA1 plate to resist deformation. The TA1 plate is pulled off when it exceeds the limited strength of the material itself.

Hill 48 quadratic function was used to describe the yield behavior of TA1 titanium plates. The anisotropy coefficients of Hill 48 (*F*, *G*, *H*, *L*, *N*, *M*) determine the tensile yield stress and shear yield stress of the material in three directions.
(1)$$\{\begin{array}{c} F=\frac{1}{2}\left(\frac{1}{R_{22}^{2}}+\frac{1}{R_{33}^{2}}-\frac{1}{R_{11}^{2}}\right);\;L=\frac{3}{2R_{23}^{2}}\\ G=\frac{1}{2}\left(\frac{1}{R_{11}^{2}}+\frac{1}{R_{33}^{2}}-\frac{1}{R_{22}^{2}}\right);\;M=\frac{3}{2R_{13}^{2}}\\ H=\frac{1}{2}\left(\frac{1}{R_{22}^{2}}+\frac{1}{R_{11}^{2}}-\frac{1}{R_{33}^{2}}\right);\;N=\frac{3}{2R_{12}^{2}} \end{array}$$

In the Formula (1), *R_ij_* is the anisotropic yield stress ratio. The paper follows the terminology that is conventional in sheet metal forming: 33 is perpendicular to the sheet surface, and 11 and 22 are parallel to the surface. The relationship with the thick isotropy index *r* is
(2)$$\{\begin{array}{c} R_{22}=\sqrt{\frac{r_{90}\left(r_{0}+1\right)}{r_{0}\left(r_{90}+1\right)}}\\ R_{33}=\sqrt{\frac{r_{90}\left(r_{0}+1\right)}{r_{0}+r_{90}}}\\ R_{12}=\sqrt{\frac{3r_{90}\left(r_{0}+1\right)}{\left(2r_{45}+1\right)\left(r_{0}+r_{90}\right)}} \end{array}$$

In Formula (2), r_0_, r_45_ and r_90_ are thick anisotropy indexes at 0 degree, 45 degree, and 90 degree directions. The mechanical properties of the TA1 materials are presented in Table 2. Material parameters are fed into the ANSYS/LS-DYNA software (ANSYS and LSTC, Livemore, CA, USA) to establish a constitutive model for the titanium plates. Only one process parameter is included in Table 2, and it varied during the course of the study, while all of the other parameters remained unchanged.

### 2.3. FEM Base Conditions

As presented in Figure 1 and Figure 4, a truncated-conical geometry with the shape of a 45 degree and 80 mm base diameter was considered for finite element simulation of the SPIF process. All the elements at the boundaries of the circular blank were assumed to be rigidly fixed. The process parameters used in this study are presented in Table 3. The perforated sheet is discretized using the shell element 163 with five integral points through the thickness of the sheet elements. Inside the element, both the stress and the strain are represented in a local coordinate system.

The forming tool with a hemispherical head was modeled as a rigid body and the tool was meshed by solid element 164, whilst the 1.5 mm size free grid was used for the grid division. The tool was translated according to the contour toolpath. The surface-to-surface contact was considered, and the coulomb friction coefficient was assigned as 0.1. Nodal displacement and rotation were constrained on all edges of the perforated titanium sheet in the SPIF process. All the above conditions were then fixed for all of the simulation cases considered in this study.

## 3. Incremental Forming Experiment Analysis of Perforated Titanium Sheet

### 3.1. The Varying Wall Angle Conical Frustum Experiment of Perforated Titanium Sheet

One of the main limitations of SPIF is the reduction in thickness, while the thickness of the sheet is only related to the wall angle. In the literature, the maximum wall angle (*θ_max_*) without sheet fracturing is commonly used to represent formability in SPIF.

According to the numerical simulation of the SPIF process of the perforated titanium sheet in the research, it is known that the circular holes become elliptical holes after forming. In the SPIF process, the wall thickness on both sides of the short axis along the circular aperture (the circumferential direction of the tool movement) is thinner than that of the long axis side (wall direction). Therefore, it is inferred that the most prone rupture of the mesh plate is near the short axis side of the elliptical hole [21].

Conventionally, the formability measure is determined by forming a series of frustums of cones or pyramids. The wall angle is increased in small steps until the sheet endures a maximum angle without fracture. This method leads to material waste in trial, and the testing time is long. As represented in Figure 5, Hussain et al. [29] proposed a variable wall angle cone to evaluate the formability, with the component wall angle from shallow to steep. This method can evaluate the maximum wall angle with a small number of specimens. The geometrical details of the formability test performed on the frustum of the cone with a continuously varying wall angle are shown in Figure 5a. This method makes use of a curved-line-generatrix to generate a revolved surface whose wall angle varies continuously. The parameters are opening diameter (*D*), pre-forming depth (*H*), radius of the circular arc (*R*), and vertical coordinate (*h*).

Experimental testing was carried out in an NH-SK1060 incremental forming machine (Nanjing University of Aeronautics and Astronautics, Nanjing, China). The perforated TA1 sheet was 120 mm × 120 mm × 1 mm in dimension, and it was subjected to deformation with a sliding tool made of W6Mo5Cr4V2. The TA1 parts were modeled in a commercial CAD/CAM software UG NX-8.5 (Siemens PLM Software, Plano, TX, USA), and the trajectories controlling the movement of the tool were automatically generated by the software. Each tool path consists of a series of contours generated transverse to the long axis of the model.

To minimize the surface friction between the tool and the perforated titanium sheet, the anti-wear hydraulic oil (98.5% pure MoS2 powder mixed with petroleum jelly in the proportion of 4:1) was used as lubricant.

By forming the perforated titanium sheet until the part breaks, the depth of the lowest point P at the crack is recorded, the wall angle on this point is *θ_max_*, and the vertebral busbar is a rounded arc. According to the geometric relations in Figure 2, the model lateral wall angle gradually varied from 30 degrees to 90 degrees. Using this model, when we determine the fracture depth (*h*) under a certain forming condition through the test, the limiting angle of the plate formation corresponding to the fracture point can be calculated. To reduce the number of tests, the target forming part is designed to have an angle, and the forming limit angle can be calculated by using Formula (1), which is as follows:(3)$$\theta_{\max}=\arccos\left(\frac{H-h}{R}\right)$$

According to Formula (3), the calculated theoretical forming limit angle is 63.06 degrees. Figure 6 is the perforated titanium sheet formation angle incremental forming experiment, and from the experiment, we can see that the forming limit angle is 63.5 degrees, the measured rupture depth is 4.7 mm, and the experimental results are 0.7% different, and, therefore, the experimental results are consistent with the calculation results. Furthermore, obvious shear can be observed in the circumferential direction (tool movement direction). And at the position indicated by the arrow in Figure 6, the fracture is a 45 degree shear fracture in the direction of the maximum stress, accompanied by a slight necking phenomenon. In the radial forming direction, there is no obvious shearing.

### 3.2. The Constant Wall Angle Conical Frustum Experiment of Perforated Titanium Sheet

The forming angle is from small to large in the limit experiment of the VWACF model, which means that the forming limit angle that is determined by the above test is not the absolute SPIF forming limit angle. In order to determine the absolute forming limit angle of the SPIF, it is necessary to determine the forming limit angle of the rotating type part. Under the same experimental conditions with the VWACF model, conical frustum incremental forming experiments were carried out, and the experimental results are represented in Figure 6. As can be seen in Figure 6, the test fracture after the maximum forming wall angle is 68.4 degrees in the above test. The wall angle is thus gradually reduced on the basis of 63.5 degrees, and when the wall angle is 50 degrees, the perforated titanium sheet is successfully formed, and the perforated titanium sheets in Figure 7b–f are all ruptured at different depths.

It can be found that the forming limit of the perforated titanium sheet is related to the forming angle. The measured result of the constant wall angle model is smaller than that obtained in the VWACF model, which is due to the greater forming force of the constant forming angle model than that of the VWACF model, and the perforated titanium sheet is thus more likely to be torn apart. In addition, it can be observed that the fracture occurs in the intermediate regions formed in the contour geometry. Along the long-axis side of the perforated titanium sheet, the wall thickness of the elements below the aperture increases slightly. In addition, the wall thickness at the bottom of the truncated cone is also slightly thickened. The results show that the fracture of the material is due to an excessive reduction of the wall thickness, rather than the deformation of the material exceeding the limit. In order to prevent this such fracture, the parts should be kept at a low level of deformation, or else the wall angle of the part should be limited in the structural design of the component.

### 3.3. Microhardness Test

A hardness experiment is an important index to evaluate the material properties. The analysis and determination of the hardness change in the deformation area of the perforated titanium sheet can explain the work hardening in the SPIF process. The Vickers hardness sampling point is small, so the microhardness of the parts in different areas can be obtained simply and intuitively. The microhardness experiment of TA1 is performed on the Vickers hardness tester (HXS-1000 AY, Hangzhou Yashite Instrumental Equipment Co., Ltd., Hangzhou, China). The test uses a diamond indenter at a 136-degree cone angle, the load is 300 g, the loading time is 20 s, and the microhardness is averaged from three points for each sample. Figure 8 shows the plot of the microscopic Vickers hardness distribution corresponding to the different positions of the circular truncated cone. It can be found that the perforated titanium sheet units under the tool (perpendicular to the thickness direction of it) have the largest microhardness values, and the microhardness value gradually attenuates along the thickness direction. This indicates that the inner surface strain of the perforated titanium sheet without contact with the tool is small. Under the action of the tool, the outer surface of the perforated titanium sheet is violently deformed. The reason for this is the combination of the friction caused by repeated rolling and a circumferential slip on the perforated titanium sheet surface, which not only refines the surface grains on the inner surface of the perforated titanium sheet but also increases the dislocation density. Thus, the microhardness of the surface is increased.

It can also be observed that the microhardness of the formed area with work hardening is significantly higher than that of the unformed area. Compared with the original perforated titanium sheet before forming, incremental molding increases the hardness of the inner surface and the outer surface. The material microhardness of the deformation zone is higher than that of the deformed zone, the main deformation zone is severely hardened, and the middle of the deformed zone is significantly softened. This is because in the deformation zone, under the repeated crushing of the tool, the strain variable of the perforated titanium sheet unit increases. As a result, the dislocation density inside the material continues to increase, the slip becomes increasingly serious, and the interaction between dislocations becomes more and more intense, and so it is manifested as an increase in the value of microhardness.

### 3.4. Perforated Titanium Sheet Fracture Morphology Analysis

A ductile fracture generally refers to the flexible fracture of the material because the stress exceeds the yield limit after a large number of plastic deformations. The general ductile fractures on the surface usually appear in the fiber area and the shear lip area, the fracture surface is presented as a dark gray fibrous shape, and microscopic morphology is usually accompanied by dimples. The fiber area is generally the fracture source region, and the expansion rate of crack in the fracture source region is slow. The shear lip area is always located at the edge of the fracture. The metal material was shear-torn under the action of force and eventually evolved into the fracture, and the fracture surface is relatively smooth. The ductile fracture indicates a better formability of the material, which is usually identified from the SEM images through the voids and shear dimples in the fracture area.

To investigate the fracture mechanism of the perforated titanium sheet, the SEM images of the fracture morphology of the perforated titanium sheet were shown in Figure 9, which was obtained by an experiment at the wall angle of 60 degrees. As observed in Figure 9a, the thickness of the fracture site is less than that of the initial perforated titanium sheet, and the cross section of the connection site among the pores is gradually changed from large to small, indicating an obvious macroscopic plastic deformation of the material, and the final perforated titanium sheet produced a ductile fracture. Furthermore, the inner surface with a small number of voids and tearing ridges is very smooth. In Figure 9b, when the surfaces of fracture were magnified ×600, the herring bone ridge-like pattern was observed in the microstructure.

In addition, microvoids were revealed, as seen in Figure 9c,d. When the fracture surface of the same region was magnified ×1000, more micropore-dimples can be observed in the plastic deformation of the perforated titanium sheet microstructure. Dimples are microvoids produced by the plastic deformation of the material within the microregion. The microvoids nucleate, grow, gather, and finally connect to each other, which leads to the effective support area on the material section and continues to decrease before, eventually, the sheet fracture and traces are left on the fracture surface. The fracture surface is clearly divided into two sections. The fracture near the outer surface is a shear-slip fracture with a wavy smooth image, which is related to the accumulation of stress in the surface. The middle layer is a dimple fracture. The initial dimple is small in diameter and shallow in depth, and when near the central area, the size and the depth of the dimple increase slightly. This indicates that the plasticity shows a gradient distribution along the thickness direction. There are a large number of smaller dimples at the inner surface in contact with the tool. The dimple size near the outer surface is large. From the inner surface to the outer surface––that is, the direction where the shear tear occurs––the dimples gradually grow and accumulate. A step-like slip band morphology appeared at the edge of the large dimples, which indicates that relatively higher numbers of plastic deformations were locally produced before the material fractured.

According to the size and distribution of the microstructural cavities (micropore, scatters, micro-defect), the plastic properties of the perforated titanium sheet, and the stresses acting on the microstructural cavity, the contour and the size of the dimples can be calculated. Different forms of dimples may occur simultaneously on the ruptured yield surface, which depends on the local stress and strain state at the time. The shear voids present in the shear region of the plastically deformed material are elliptical holes in the neck of the shear surface in the area of the plane stress state. During the shearing process of the material, open or closed shear voids can be formed. They extend into trajectories of stress effects, and their combination occurs in the plane of the maximum shear stress. As is shown in Figure 9d, from the fracture morphology of the outer surface area of the fracture at a magnification of 1000 in the red box, it can be observed that there are a large number of smooth planar morphology of 2 to 5 μm and shallow small dimples. These dimples are interwoven and clustered together, which makes the crystal boundary weaken and leads to the transgranular fracture. This will only occur when the axial force on the perforated titanium sheet is greater than the shear force, though, which is generated by the tool.

## 4. The Force Trend in SPIF of Perforated Titanium Sheet

The deformation takes place between the tool and the perforated titanium sheet during the SPIF process, and the forming forces have a great impact on the accuracy and the forming mechanism of the perforated titanium sheet. A series of calculations and tests show that the forming force is a function of the following main influencing factors:(4)$$F=f(t_{0},\sigma_{s},\psi_{t},s,r_{p},\phi ,\Delta )$$

In Formula (4), *t_0_* is the initial thickness of the plate (mm), *σ_s_* is the yield limit of the material, *Ψ_t_* is the thinning rate of the plate, s is the step-down size (mm), *r_p_* is the tool radius (mm), *φ* is the half cone angle, and Δ is the degree of deviation from the sine law (mm). The forces corresponding to the Cartesian coordinates are shown in Figure 10, which depicts the forming force components associated with the *X*-, *Y*-, and *Z*-axes in the SPIF process of the perforated titanium sheet. The total deformation force vector can be decomposed into the axial (*F_Z_*) and horizontal (*F_XY_*) components. The horizontal component *F_XY_* can be decomposed into a component in the OX direction (represented by *F_X_*) and the components in the OY direction (represented by *F_Y_*).

Figure 11 compares characteristics of the forming force associated with the *X*-, *Y*- and *Z*-axis. The forming process was to form the truncated cone shape with 80 mm in diameter, and 10 mm in depth and feed velocity remained constant at 1000 mm/min. When the tool moves along the m-direction of the contour circumferential trajectory, it will produce a steadily growing axial force (*F_Z_*) and two horizontal forces (*F_X_* and *F_Y_*) in the form of sinusoidal fluctuations. As shown in Figure 10, the tool moves along the m direction and returns to the starting point (*b*) before the subsequent round of molding begins (*b*→*b*). Then the tool lifts to the point (*c_lift_*) and moves down along *Z*-axis (n direction) to point (*c_down_*), where the procedure starts over again.

(1)Relative to a contour line, each increment of all three force components have peaks and valleys. *F_X_* and *F_Y_* change alternately in a sinusoidal pattern with a phase difference of π/2, which is due to the anisotropy of the material and asymmetric deformation modes. Each time the axial force *F_Z_* shows a significant increase, the corresponding values of horizontal forces (*F_X_* and *F_Y_*) are nearly 0, and this is due to the fact that horizontal forces have not yet started when the tool does the step leap between layers.(2)It is obvious that *F_Z_* is the main cause of deformation force, and the axial force is therefore an important component in the later test. When the tool moves along the same contour (*b*→*b*), *F_Z_* is relatively constant, and the force curve fluctuates gradually as the tool passes through each hole. A fixed depth increment (step-down size *s*) was applied between the successive contours. Upon completing one round contour, the forces associated with the three axes (*F_Z_*, *F_X_*, and *F_Y_*) were are all decreased to 0 during the interval when the tool was lifted (point *c_lift_*). When the tool was again pressed onto the perforated titanium sheet down the n direction to point *c_dow_*_n_, the *F_Z_* curve shows a sharp pulse, and the maximum *F_Z_* increases again and reaches a peak of the pulses. The tool will then continue to proceed along the predetermined path in the next round (*d*→*d*). The trend of forming forces increased with greater depth in the *Z* axis and tended to stabilize after reaching a certain depth.

**Figure 11 materials-16-03176-f011:**
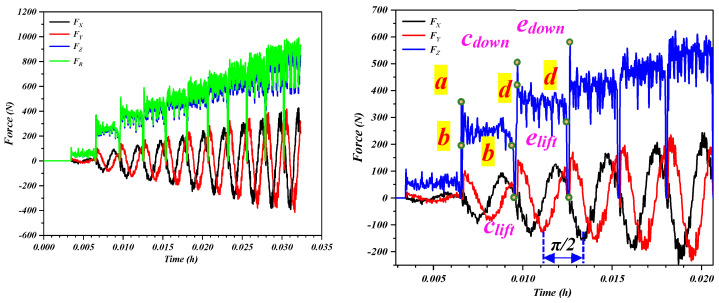
Three-axis forces for the truncated cone of wall angle 45 degree with 1 mm sheet thickness, 10 mm forming depth, z-level tool path with 1 mm step-down size, and 10 mm tool diameter for perforated titanium sheet; point (b) and (d) denotes the start point of the tool path in the corresponding layer, once a complete circuit has been made, back to point (b) or (d), the tool lifts to point (*c_lift_*) or (*e_lift_*) and then moves down by n direction to point (*c_down_*) or (*e_down_*).)

Figure 12 represents the evolution of the three force components by generating a cone with standard process parameters. The tool path used in this part of the analysis follows a continuous trajectory. Since the contact and separation of the tool with the perforated titanium sheet occur continuously, the oscillatory forces are observed. The force trend is mainly attributed to the bending effect. In terms of the *F_Z_* variation, most of the whole typical forming force curve could be divided into four distinct regions: *A*, *B*, *C*, and *D*.

At the early stages of the SPIF process, there is a certain gap between the deformation area and the clamp, and the extrusion of the tool leads to the bending of the perforated titanium sheet. In addition, the tool has not formed a stable contact area until a large number of contour is formed, and, furthermore, the deformation of the perforated titanium sheet must overcome the influence caused by the edge of the backing plate. The support backing plate applies to the perforated titanium sheet an initial forming mechanism that can be described as bending, which is gradually shifted as the nominal forming angle is reached in the steady-state incremental formation state. It is worth noting that the *F_Z_* curve of the perforated titanium sheet has a large mutational amplitude in the early stages of the SPIF process (region *A*), but the overall trend is slightly upward. This is due to the poor stiffness and large elastic deflection at that time. In addition, the perforated titanium sheet does not form enough contour for the insufficient forming depth, and the forming force does not fully enter the stable state.

As the perforated titanium sheet was gradually processed into three-dimensional shaped parts by the SPIF process, the wide deformation of the entire perforated titanium sheet changed to a highly localized deformation of the metallic material under the perforated titanium sheet area in contact with the tool, and the stiffness and the maximum forming forces increased significantly. This is the reason why the increased rigidity of the perforated titanium sheet during the SPIF process allows the perforated titanium sheet to resist the widely distributed deformation, and the increased rigidity allows the perforated titanium sheet to resist the widely distributed deformation, too, and also limits the deformation to the area that the tool contacts. In addition, according to the characteristics of the stress-strain curve, the initial development of plastic deformation leads to the working hardening of the material. When each unit of the perforated titanium sheet deforms gradually, the strain hardening effect of the unit increases. Thus, it can be observed that force *F_Z_* begins to rise gradually in region *B* until it peaks.

However, further deformation after the peak may lead to the excessive thinning of the material that reduces the magnitude of the forming force. Thus, the posterior contour of the forming force is determined by the combined action of the perforated titanium sheet strain hardening (which causes increased force) and thinning (which tends to reduce the required force). As shown in Figure 10, in the first case, the *F_Z_* curve trend in region *C* shows a low negative gradient in the curve after the peak appears, which is due to the serious thinning of the perforated titanium sheet in the previous stretching stage, and the strain rate under the incremental nature decreases, so that the decrease in the forming force decreases. The second case is that after the peak appearance of region *B*, the forming force in region *C* enters a stable state, which is due to the thinning effect and the strain hardening effect of the perforated titanium sheet compensating for each other. Therefore, the curve gradient after the peak can serve as a critical indicator of the studied SPIF process. The third case is that the curve of the axial forming force decreases the monotonic linearly due to the serious thinning of the perforated titanium sheet. In conclusion, the curvilinear gradient after the peak in region *B* can serve as a key indicator for the SPIF process that was studied on the perforated titanium sheet.

In some cases, due to the resumption of the interaction between stretching (thinning) and strain hardening, it can be observed that there are several millimeters of depth on region *D*. The rebound growth of the forming force indicates that the strain hardening effect of the perforated titanium sheet is stronger than the stretching effect, and so it is manifested as a small increase in the forming force again. This can also be explained by the bottom of the material and the accumulation of the material hardening interaction, the deepening of the forming depth of the material sliding becoming more difficult and eventually leading to the deformation resistance of the titanium plate, and the increase in the perforated titanium sheet, all of which lead to the performance of the forming force increasing. 

## 5. The Influence of Wall Angle on Perforated Titanium Sheet Forming

The simulation results and the actual results of the forming parts are shown in Figure 13. The initial thickness of the perforated titanium sheet is 1 mm. It can be clearly observed insofar that the error of the experiment and simulated values of the wall angle do not exceed 5%. Therefore, the numerical simulation basically agrees with the experimental results.

In order to study the influence of different wall angles on the formation of the perforated titanium sheet at the same forming depth, and in order to provide process guidance for the formation of components, the finite element model was used to simulate the SPIF process of workpieces with different wall angles. The standard values used to conduct a series of experiments of the TA1 perforated titanium sheet are: the upper surface radius of the target sample as of 40 mm, the tool diameter (*dp*) as of 10 mm, the step-down size (s) as of 0.5 mm, the initial thickness (*t_0_*) as of 1.0 mm, and the forming depth (*h*) as of 15 mm. Under the same conditions as other process conditions, the formation process of the perforated titanium sheet was to form the truncated cone at the wall angles of 25, 30, 35, 40, 45 and, 50 degrees.

### 5.1. The Influence on Forming Stress

Continuous deformations will produce more dislocations, which interact and create voids that eventually lead to fracture. The ultimate strain before failure depends on the stress state: a high level of compressive stress will squeeze the voids and slow down the development of the damage [24]. This is why shaping processes with mainly natural compression, such as rolling and wire drawing, can produce large amounts of strain in the material without causing damage. This is in contrast to the process of operating mainly during tension.

As represented in Figure 14, when the wall angle increases from 25 degrees to 35 degrees, the principal stresses (the tensile stress and the compressive stress) of the perforated titanium sheet showed an increasing trend. The deformation area has a certain gap from the clamp, and the extrusion of the tool leads to the perforated titanium sheet material bending, with the bending dominating the initial stage of SPIF process, and with the tensile thinning of the material not significant for the small wall angle. Therefore, it can be found that the values of the effective stress differ less.

When the wall angle increases from 35 degrees to 40 degrees, the absolute values of the principal stresses all increase rapidly, and the significance rises in both stretching and bending with more severe plastic deformation, which means that the effective stress is also reflected in the rapid growth. The above phenomena can be explained by the fact that the trajectory of the deformation area of the tool in contact with the perforated titanium sheet is a closed ring band. The latter layer of the forming belt partially coincides with the former layer, and the coincidence part has been deformed in the previous layer. However, as the wall angle increases, there is larger portion of hardened material that is deformed by the tool in each interlayer movement between continuous tool paths. Thus, the stress required for the deformation increases.

When increasing from 40 degrees to 45 degrees, both tensile stress and compressive stress decreased slightly, and the effective stress of 45 degrees was less different than the equivalent effect of 40 degrees. As the wall angle continues to increase to 50 degrees, the absolute values of the principal stresses and effective stress are observably reduced, and due to the lower stress state caused in the material of the perforated titanium sheet, the formability of the plate increases and the vertical direction force (*F_Z_*) decreases.

### 5.2. Influence on Forming Strain

The simulation results of the deformation strain of the perforated titanium sheet element are represented in Figure 15, and it can be seen from Figure 15 below that at the end of forming, as the wall angle increases from 25 degrees to 50 degrees, the maximum value of the effective plastic strain increases significantly, the maximum values of the first principle strain and the effective strain grow as an exponential function, and, finally, the third principal strain shows a secondary polynomial growth. The effective plastic strain increases from 0.57 to 2.28. The deformation of each unit is gradually increasing in the SPIF process, yet within each contour, the deformation pattern does not actually increase monotonically, and as the strain path changes, radial elongation occurs within the contour and the thickness thins in that direction.

The pairs of principal strain and equivalence changes along the *X* axis with different wall angles are represented in Figure 16, from which it can be seen that the deformation of each element increases gradually as the SPIF process progresses. However, within each contour of the perforated sheet, the deformation mode is not actually monotonic, so the strain path changes.

(1)Radial elongation and thinning in the thickness direction are obvious in the contour, so the strain level of the large wall angle is higher than that of the small inclination.(2)The change in strain paths within each contour leads to an accumulation of deformation, and they repeat the contour trajectory until the tool moves away from the point. This deformation then reaches a maximum value. Thus, the accumulated effective plastic strain becomes different from the effective strain obtained directly from the strain component. With the change of the wall angle, the maximum value of the effective plastic strain increases with the increase in the wall angle, and in the main deformation zone, the maximum value ε_p_ of the effective plastic strain is almost all located on both short axis sides of the circular aperture (the circumferential direction of the tool movement), which is located in the discontinuous region of the perforated titanium sheet. The deformation mechanism of SPIF in these regions can be explained as the phenomenon of stretching and shear in the plane perpendicular to the movement direction of the tool, while the shear is in the plane parallel to the movement direction of the tool.

**Figure 16 materials-16-03176-f016:**
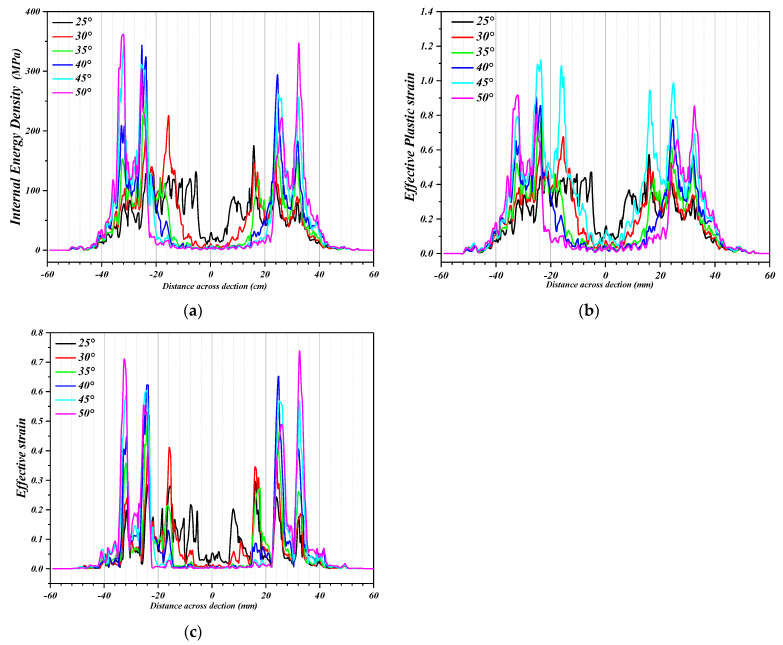
Comparison diagram regarding the principal strain and effective strain along *X*-Axis (Y = 0 cross section) with different wall angles. (**a**) The internal energy density curves along *X*-Axis (Y = 0 cross section) with different wall angles. (**b**) The effective plastic strain curves along *X*-Axis (Y = 0 cross section) with different wall angles. (**c**) The effective strain curves along *X*-Axis (Y = 0 cross section) with different wall angles.

### 5.3. Influence on the Median Surface Contour Accuracy

When the wall angle is small, it can be considered that the perforated titanium sheet only occurs during pure stretching, without any obvious through-thickness shear. When the wall angle increases, the total deformation of the perforated titanium sheet consisted of an in-plane stretching and a shear along the thickness direction. As can be observed from Figure 17, in the early stages of the SPIF process, the perforated titanium sheet mainly shifts along the Z-direction. The units at the bottom of the cone undergo a rigid translation, namely, the tool pushes the radial outward material into the cone wall. However, as the SPIF process continues, the radius of the tool contour is smaller, and the material at the bottom of the truncation cone is pushed to the cone wall due to plastic deformation, and a small radial-outward displacement can be seen.

The units at and around the hole (the discontinuous region) appear in the inverted-concave form (the normal error h2), while the units away from the hole (continuous region) appear in the outward-convex form (the normal error h1), and both the normal error h2 (concave) and h1 (convex) in the side wall of the perforated titanium sheet gradually decrease with the increase in the wall angle of the perforated titanium sheet. That is to say, with the increase oin the wall angle, the out-of-plane deformation produced by the continuous region of the perforated titanium sheet and the in-plane deformation of the discontinuous region are effectively alleviated. Furthermore, the deformation degree at the tool point (on the left contour of Figure 17) is obviously larger than that of the side wall of the cone table, which is at the same height. Due to this increased deformation, an obvious indentation band is often formed at the tool point.

Figure 18 can be observed that the maximum offset in the X and Y directions is relatively consistent, and the disparity in the minimum offset of X and Y is due to the design of the machining trajectory. At the wall angle of 25 degrees to 35 degrees, the maximum offset deviation in the X and Y directions decreases with the increase in the wall angle, especially in the small wall angle molding, and due to the fact that the rebound effect of the material and the forming depth is not large, the contour deviation is large. When the wall angle continues to increase to *45* degree, the maximum offset deviation in the X and Y directions increases slightly due to the material properties of the perforated titanium sheet.

### 5.4. Influence on Wall Thickness and Thinning Ratio

Under the high wall Angle of SPIF, the elongation of the material in the perforated titanium sheet deformation area increases and the deformation of the element increases, and so the strain value increases and the fluctuation of the wall thickness curve increases. The macroscopic performance is that the maximum thinning rate of the forming parts increases.

Figure 19 shows the wall thickness of the perforated titanium sheet in different wall angle conditions. Figure 17 shows the thickness contour and theoretical sinusoidal thickness and radial dimension of the perforated titanium sheet at different wall angles along the *X* axis at the end of forming. The wall thickness of perforated titanium sheet at 25, 30, 35, 40, 45, and 50 degree wall angles estimated by the sine law can be easily calculated to be 0.91 mm, 0.87 mm, 0.82 mm, 0.77 mm, 0.71 mm, and 0.64 mm, respectively. As depicted in the cloud map of Figure 16, it can observe that the distribution zones of the wall thickness were between (0.8937–0.9259) mm, (0.843–0.877) mm, (0.8033–0.8418) mm, (0.7688–0.8195) mm, (0.7479–0.8081) mm, and (0.7202–0.7822) mm, respectively. The relative errors were −1.79 percent to −1.75 percent, −3.1 percent to −0.8 percent, −2.04 percent to −2.66 percent, −0.16 percent to −6.43 percent, −5.34 percent to −13.82 percent and −12.53 percent to −22.22 percent, respectively. As the wall angle increases, the relative error of thickness gradually increased.

As can be seen in Figure 20, the thickness of the perforated titanium sheet with different wall angles is lower than the thickness predicted by the sine law at the thinnest point, which means that the perforated titanium sheet thins along the cone wall. Indeed, in the SPIF process, the perforated titanium sheet is severely stretched along the radial direction, and is stretched along the circumferential direction under the plane strain condition. At the same time, the shell thickness of the perforated titanium sheet is seriously reduced, and the field distribution of the thickness varies with the wall angle. As the wall angle increases, a fracture occurs when the thickness decreases to a threshold. By increasing the wall angle, a continuous, intermediate shape can be formed, which can contribute to a significant expansion of the process window. It was observed that in the plane perpendicular to the direction of the forming trajectory, the deformation of the perforated titanium sheet is mainly caused by stretching and bending. In the plane parallel to the direction of the forming trajectory, a significant thickness shear was observed. In the main deformation area, the thickness fluctuates dramatically in the noncontinuous region (hole domain area), but fluctuates less in the continuous region (areas outside the hole). The perforated titanium sheet has a slow thinning process at first, and then gradually increases, but in the end it tends to a stable value; the thinning is mainly distributed in both of the short axis side of the aperture. The bottom of the part is mostly not deformed and still has a close thickness to the initial value (1 mm), this also indicates that little plastic deformation occurs. The differences between them may be due to the different deformation properties.

Figure 21 is the cloud diagram of the perforated titanium sheet thinning rate with different wall angles. The thinning region of perforated titanium sheet shown in the figure is a property of the SPIF process and not a property of a specific geometry. The perforated titanium sheet will be drastically thinned at the large wall angle, and both sides of the short axis of the perforated titanium sheet circular hole are red and orange on the thinning rate cloud—that is, the place where the thinning rate is large, and the long axis shows green and blue in a small area, indicating that the perforated titanium sheet matrix material shows a slight thickening trend in a small area along the long axis. The increased wall angle increases the uneven thinning of the perforated titanium plate.

As represented in Figure 22, the increase in the wall angle directly leads to a thinning of the interface. With the increase in wall angle from 25 degree to 35 degree, the minimum thickness of the main deformation area of the assembly decreases linearly, and the perforated titanium sheet conical table parts of different sizes can be formed. When the wall angle moves up to 45 degree, the thickness distribution of the perforated titanium sheet is more uniform. However, as the wall angle continued to increase, the relative error of thickness gradually increased, the minimum thickness of the forming area decreased significantly, the formability of the perforated titanium sheet decreased, and the uniformity of the sheet thickness deteriorated. Therefore, the appropriate wall angle should be selected in the actual processing process.

### 5.5. Influence on Forming Force

Figure 23 compares the evolution histories of the vertical force vector *F_Z_* at the varying of wall angle. When the wall angles varies between 25 degrees and 30 degrees in the SPIF process, due to the insufficient forming depth, the perforated titanium sheet has not formed enough contour, the forming force has not fully entered the stable state, and the forming force fluctuates greatly, which will cause the vibration of the perforated titanium sheet, which affects the accuracy of the manufacturing parts. The results are consistent with the normal errors, which are observed in Figure 17, and the tensile thinning of the material is not significant for the small wall angle. The perforated titanium sheet has experienced a much longer bending process before the strain hardening, which delays the realization of stable conditions.

With the continuous increase in the wall angle, the forming force enters a stable state, and the problem of large fluctuation of the forming force is alleviated. When the wall angles varies between 35 degrees and 45 degrees in the SPIF process, it can be observed that the axial force becomes uniform. In addition, the axial peak forces *F_Z_* (in region B) increased significantly as the wall angle continues to increase. To obtain higher wall angles, the contact area between the tool and the perforated titanium sheet decreases, which makes the amount of material available for local deformation in a single contour decrease at any instant in the forming direction, but the perforated titanium sheet can obtain a better sheet metal support, so the peak of the axial forming force *F_Z_* in region *B* gradually increases.

The friction has always existed in the sliding and rolling contact between the tool and the perforated titanium sheet. The friction heat will be concentrated at and around the contact unit with the tool. The temperature rise, generated by the friction heat, will provide a certain amount of energy for the sliding system composed of the tool and the perforated titanium sheet. The energy can activate more sliding planes to get more plastic deformation and promote the dislocation movement. In addition, the thinning of the perforated titanium sheet plays the most relevant role and the strain hardening cannot permit the achievement of equilibrium conditions. As a result, it was observed that the forming force is reduced to varying degrees after peaks (in region *C*), and the range of region *C* also increases slightly with the forming angle.

When the forming wall angle continues to increase to 50 degree, the axial peak force *F_Z_* decreases. In addition, with the wall angle continues to increase from 40 degrees to 45 degrees, the stiffness of the perforated titanium sheet is reduced, and the maximum of forming force also shows a trend of increase in region *D*. The thinning rate of the perforated titanium sheet increases linearly with the forming angle, and when the wall angle continues to increase to 50 degrees, it can be seen that that the peak of forming force in region *B* is decreases, which can be explained by the fact that the serious thinning in the previous stretching stage, and the strain rate under the incremental nature decreases. Therefore, the material work hardening cannot compensate sheet thinning, and, moreover, due to the lower stress state caused in the material of perforated titanium sheet, the formability of the plate increases and the vertical direction force (*F_Z_*) decreases.

Under the continuous extrusion and friction of the titanium plate and the perforated titanium sheet, the material flows along the circumferential (forming) direction and produces the material accumulation phenomenon. The friction occurs in the upper surface area where the tool and the plate contact, so the friction in the circumferential direction will be more significant than the wall direction and the thickness direction. As is present in Figure 23 and Figure 24, the friction energy increases as the wall angle increases, and the forming forces in the SPIF process are all increased. The friction caused by the circumferential movement of the tool is the largest, the growth rate of the horizontal forming forces (*F_X_* and *F_Y_*) are significantly higher than the axial forming force (*F_Z_*).

## 6. Conclusions

In this paper, the effects of the wall angle range, fracture mechanism, and wall angle of TA1 perforated titanium sheet incremental forming on the forming stress strain, contour accuracy, wall thickness thinning, and forming forces of the perforated titanium sheet components were studied through experiments and finite element simulation, and the results obtained were as follows:(1)Through the forming limit angle experiment of the variable wall angle cone model (VWACF), the forming limit angle is 63.5 degrees, and the theoretical forming wall angle of 63.06 is 0.7% different from the experimental results, and the experimental results are consistent with the calculation results. Shear fracture occurs on both sides of the short axis, and the fracture and maximum stress direction are 45 degrees, accompanied by a slight neck shrinkage. The limit angle of the perforated titanium sheet forming obtained by the constant forming angle experiment is 50 degrees. By observing the fracture morphology, the fracture mode of the perforated titanium sheet in the incremental forming process is the ductile fracture.(2)The growth rate of the horizontal forming forces are significantly higher than the axial forming force. In terms of the *F_Z_* variation, most of the typical forming force curve could be divided into four distinct regions: *A*, *B*, *C*, and *D*. The force trend of TA1 perforated titanium sheet can be attributed to the bending effect of the sheet in the early phase of the process (region *A*) and the combined effects of sheet thinning and strain hardening in the second stage of the process (region *B*). The curve gradient after the peak can serve as a critical indicator of the SPIF process of the perforated titanium sheet.(3)The forming wall angle has the greatest influence on the strain, wall thickness, and geometric accuracy of the perforated titanium sheet components. The work-piece with the small wall angle possesses relatively small stress strain and shell thinning rate, but a relatively large contour deviation. The effective strain, forming forces, and friction energy increase as the wall angle increases. As the wall angle increases to 45 degrees, less thinning and a more homogenous plastic strain and thickness distribution were achieved and the geometric accuracy is ideal.

## Figures and Tables

**Figure 1 materials-16-03176-f001:**
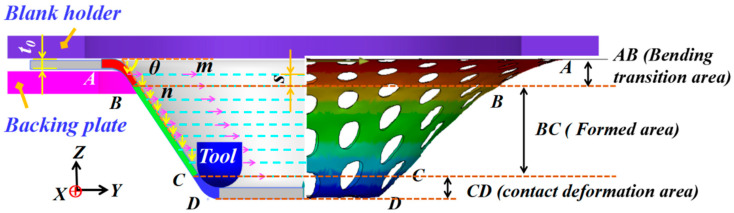
Schematic Diagram of Incremental Forming Principle. (Initial plate thickness t_0_, wall angle *θ*, and step down size *s*.)

**Figure 2 materials-16-03176-f002:**
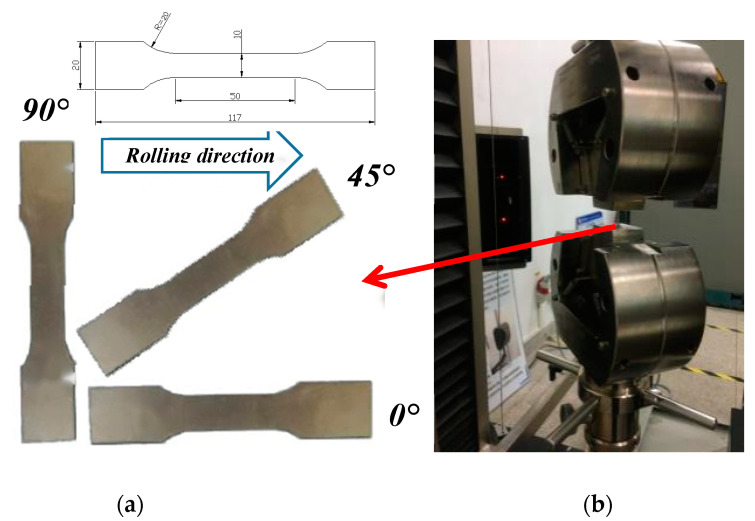
Specimen of TA1 plate for uniaxial tensile test. (**a**) Specimen shape and location for tensile test (Unit: mm). (**b**) Tensile testing machine.

**Figure 3 materials-16-03176-f003:**
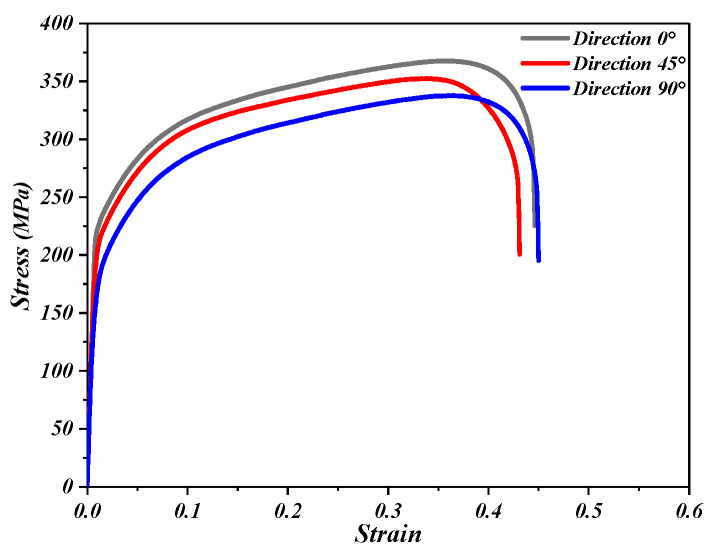
Experimentally measured true stress-strain curves of TA1.

**Figure 4 materials-16-03176-f004:**
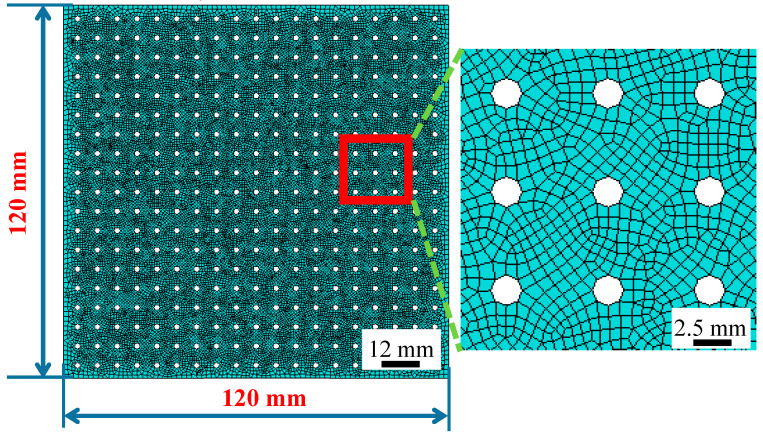
The Schematic representation of the grid partition of the perforated titanium sheet.

**Figure 5 materials-16-03176-f005:**
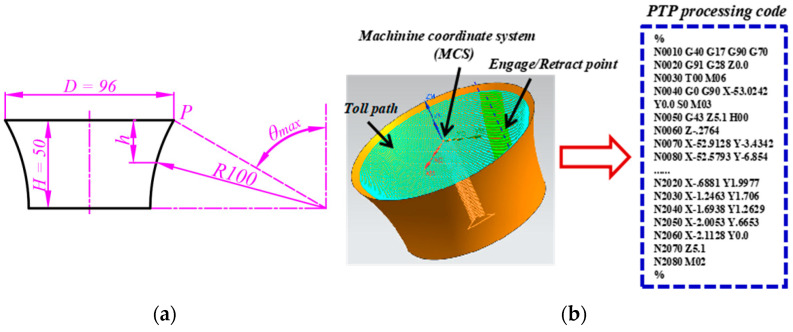
The varying wall angle conical frustum (VWACF) test modal of SPIF. (**a**) Illustration model of the test (mm). (**b**) Schematic diagram for the toolpath.

**Figure 6 materials-16-03176-f006:**
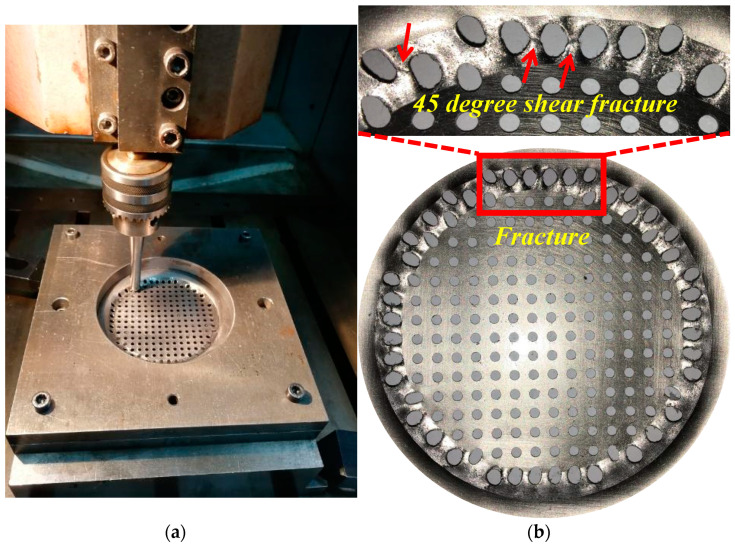
The perforated titanium sheet forming for changing forming angle model. (**a**)The torn formed modal is mesure to find the deepest value or the maximum deformable an-gle. (**b**) Fracture position.

**Figure 7 materials-16-03176-f007:**
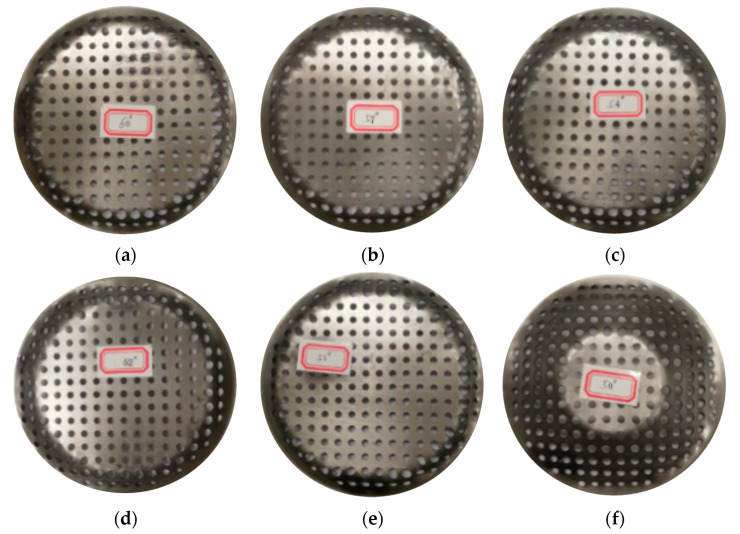
The perforated titanium sheet formed by different constant wall angle model. (**a**) Wall angle of 60 degrees (fracture). (**b**) Wall angle of 57 degrees (fracture). (**c**) Wall angle of 54 degrees (fracture). (**d**) Wall angle of 52 degrees (fracture). (**e**) Wall angle of 51 degrees (fracture). (**f**) Wall angle of 60 degrees (succeed).

**Figure 8 materials-16-03176-f008:**
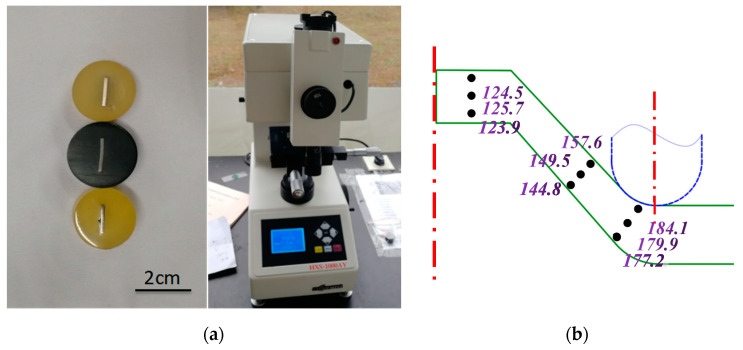
The microhardness test result of circular truncated cone after SPIF of the perforated TA1 sheet. (**a**) Microhardness test samples and equipment. (**b**) Microhardness distribution in workpiece cross-section of perforated TA1 sheet (HV).

**Figure 9 materials-16-03176-f009:**
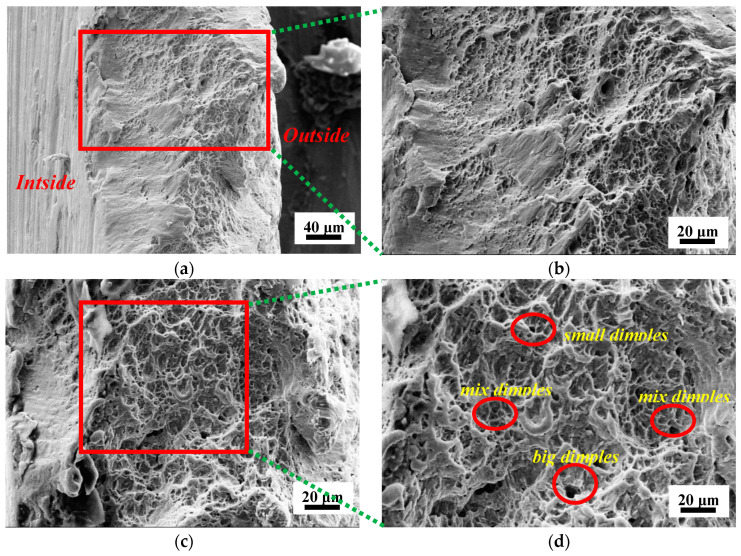
The SEM fracture scan imagery for perforated titanium sheet sample with wall angle of 60 degrees. (**a**) Overall scan diagram of the fracture. (**b**) The magnification diagram near the inner surface of the fracture. (**c**) The magnification diagram near the outer surface of the fracture. (**d**) The fracture morphology of the outer surface area of the fracture at magnification of 1000 in the red box.

**Figure 10 materials-16-03176-f010:**
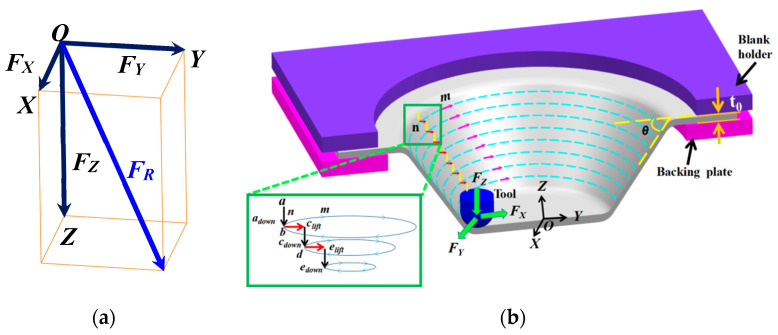
Synthesis of the forming force and the movement route of the tool. (**a**) Force componenyts in a Cartesian coordinate system. (**b**) Cross-section view of SPIF process and its parameters. Point (b) and (d) denotes the start point of the tool path in the corresponding layer, once a complete circuit has been made, back to point (b) or (d), the tool lifts to point (*c_lift_*) or (*e_lift_*) and then moves down by n direction to point (*c_down_*) or (*e_down_*).

**Figure 12 materials-16-03176-f012:**
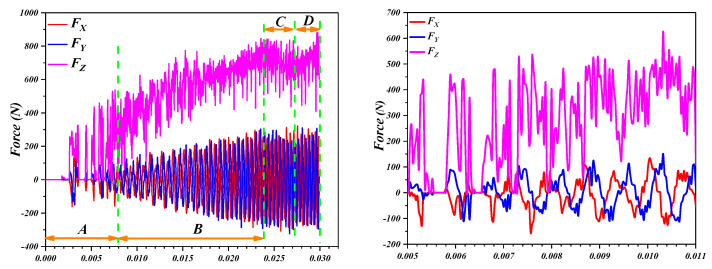
Three-axis forces for the truncated cone of the wall angle of 45 degrees, with 1 mm sheet thickness, 25 mm forming depth, z-level tool path with 0.2 mm step-down size, and 10 mm tool diameter for perforated titanium sheet.

**Figure 13 materials-16-03176-f013:**
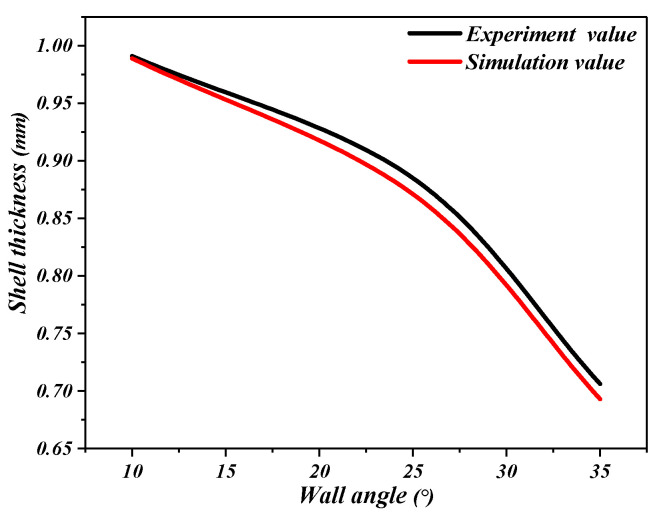
Wall thickness comparison of the experimental and the numerical simulation.

**Figure 14 materials-16-03176-f014:**
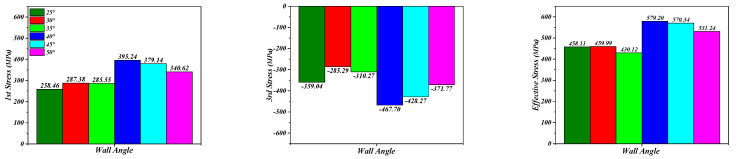
Comparison diagram regarding the maximum principal stress and effective stress with different wall angles.

**Figure 15 materials-16-03176-f015:**
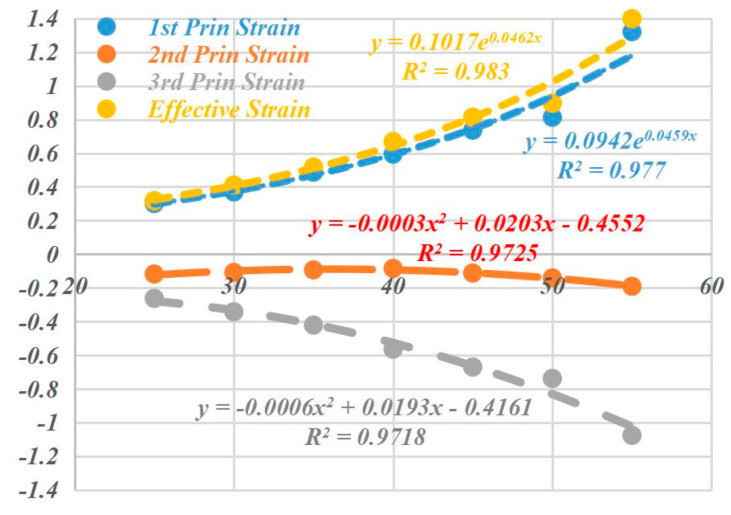
Extreme value changing curve regarding the principal strain and effective strain of different wall angles.

**Figure 17 materials-16-03176-f017:**
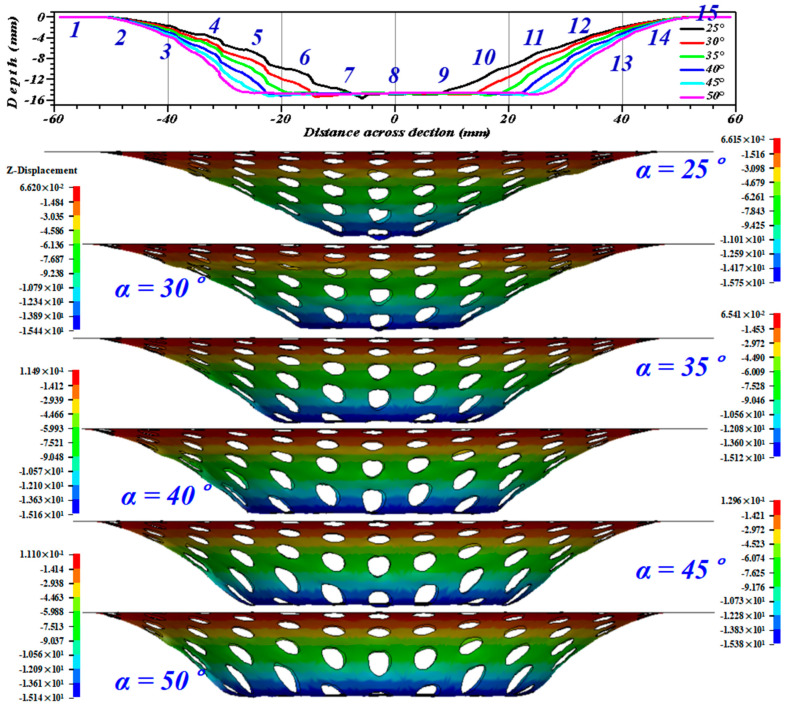
The outline of the perforated titanium sheet truncated-cone workpiece with different wall angles along the *X*-axis (Y = 0 cross section).

**Figure 18 materials-16-03176-f018:**
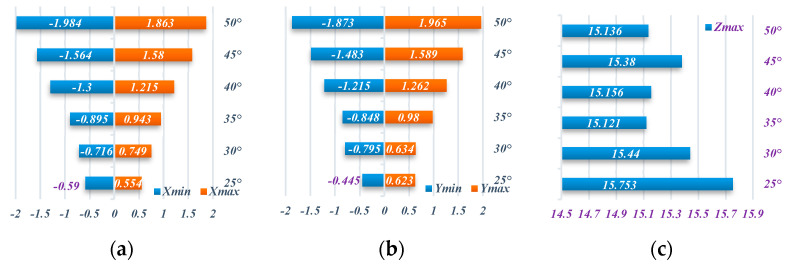
The maximum displacement of the perforated titanium sheet truncated-cone with different wall angles. (**a**) The maximum X-displacement. (**b**) The maximum Y-displacement. (**c**) The maximum Z-displacement.

**Figure 19 materials-16-03176-f019:**
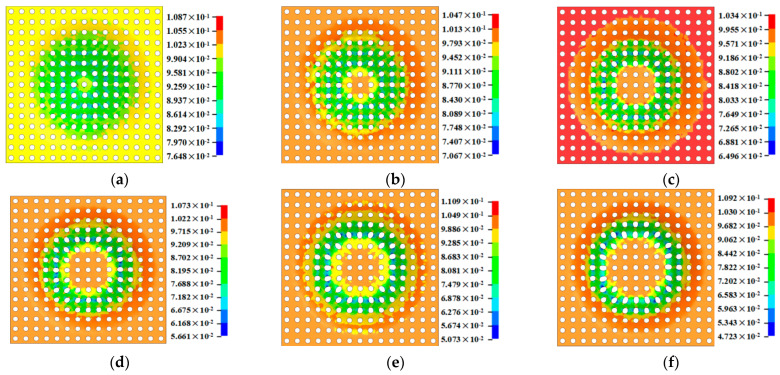
Cloud image of perforated titanium sheet wall thickness with different wall angles. (**a**) Cloud image of wall thickness at 25 degree. (**b**) Cloud image of wall thickness at 30 degree. (**c**) Cloud image of wall thickness at 35 degree. (**d**) Cloud image of wall thickness at 40 degree. (**e**) Cloud image of wall thickness at 45 degree. (**f**) Cloud image of wall thickness at 50 degree.

**Figure 20 materials-16-03176-f020:**
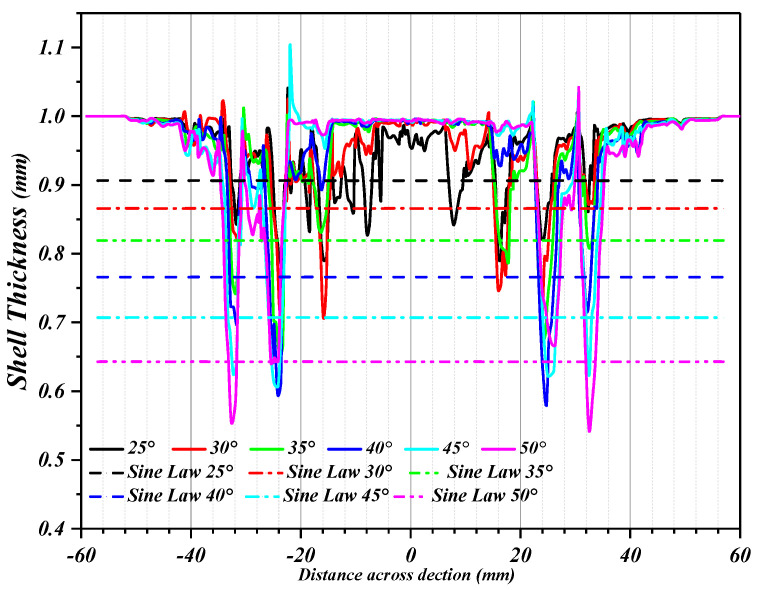
Wall thickness curves of the perforated titanium sheet along *X*-axis (Y = 0 cross section).

**Figure 21 materials-16-03176-f021:**
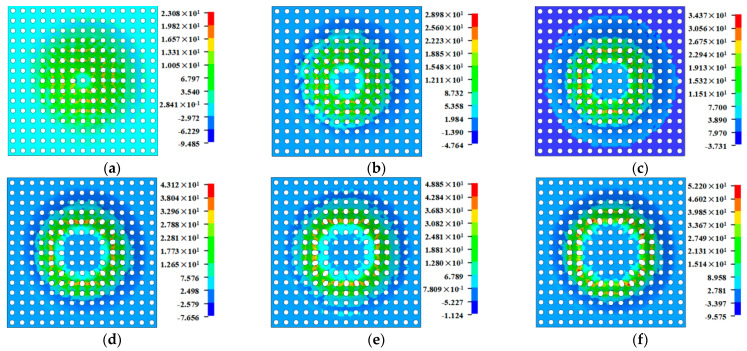
Cloud image of perforated titanium sheet thinning ratio with different wall angles. (**a**) Cloud image of thinning ratio at 25 degree. (**b**) Cloud image of thinning rate at 30 degree. (**c**) Cloud image of thinning ratio at 35 degree. (**d**) Cloud image of thinning rate at 40 degree. (**e**) Cloud map of thinning ratio at 45 degree. (**f**) Cloud map of thinning rate at 50 degree.

**Figure 22 materials-16-03176-f022:**
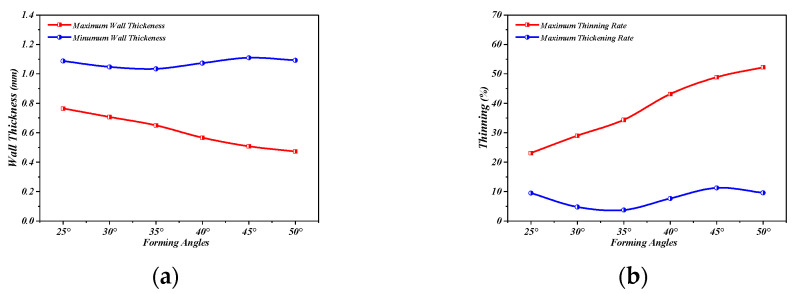
Trend diagram of wall thickness and thinning rate of perforated titanium sheet with wall angle change. (**a**) Trend diagram of the maximum wall thickness at different wall angles. (**b**) Trend diagram of the maximum thinning rate at different wall angles.

**Figure 23 materials-16-03176-f023:**
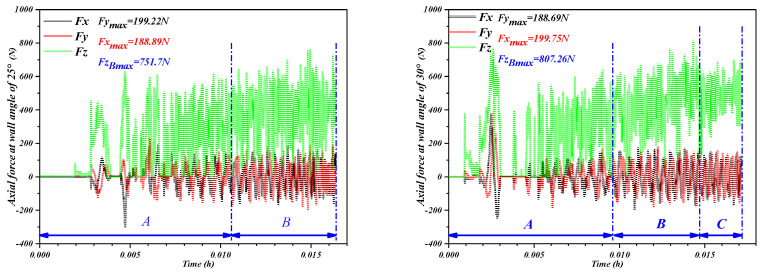
The trend of horizontal force and vertical force with the varying wall angle of perforated titanium sheet. (A, B, C and D are the lengths of regions A, B, C and D, respectively).

**Figure 24 materials-16-03176-f024:**
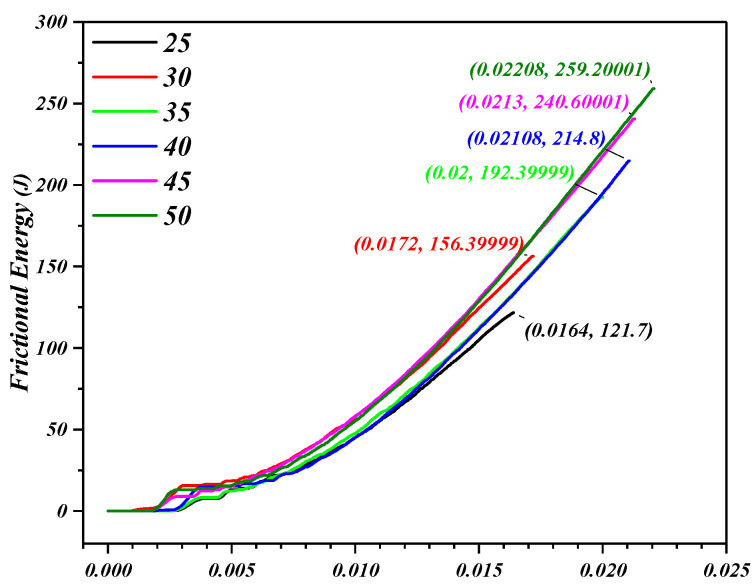
Frictional energy trend with the varying wall angle of perforated titanium sheet.

**Table 1 materials-16-03176-t001:** Chemical composition of pure titanium TA1 (%, mass fraction).

Fe	C	N	H	O	Ti
≤0.2	≤0.08	≤0.03	≤0.015	≤0.18	the rest

**Table 2 materials-16-03176-t002:** Material parameters of TA1.

Density Kg·m^−3^	Modulus of Elasticity *E*/GPa	Poisson Ratio *v*	Yield Strength *R_p0.2_*/MPa	Tensile Strength *R_m_*/MPa	Hardening Coefficient *n*	Resistance Coefficient *K*/MPa	Thick Anisotropy Index *r*
4504	109.68	0.34	236.7	341.2	0.1294	520.11	1.697

**Table 3 materials-16-03176-t003:** Perforated titanium sheet incremental forming process parameters.

Parameter Name	Initial Wall Thickness *t_0_*/mm	Tool Diameter *dp*/mm	Wall Angle *θ*/°	Feed Velocity *v_2_*/(mm/min)	Step-Down Size *s*/mm	Forming Depth *h*/mm	Aperture Diameter/mm	Hole Spacing/mm
Numeric value	1	10	45	1000	0.2	25	4	8

## Data Availability

Not applicable.

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
