# Peer review of "The Formability of Perforated TA1 Sheet in Single Point Incremental Forming"

_materials, 2023, doi:10.3390/ma16083176_

Round 1

Reviewer 1 Report

This paper studies the wall angle range of TA1 titanium mesh incremental forming by combining experiments and finite element simulation, analyzes the fracture mechanism by analyzing the fracture morphology.

In order to improve the quality of your paper, major points should be addressed and corrected:

1.      At first point of view, similar article could be seen in literature, published before by authors (Ref [23]: Li, R.X.; Wang, T. Research on Single Point Incremental Forming Characteristics of Perforated TA1 Sheet[J]. Metals. 2022, 12, 1944) . Therefore, the novelty is unclear.  The novelty of this article should be emphasized and should be addressed in the introduction regarding what you have developed, found, or improved compared with the other similar research’s published before by authors.

2.      It would be necessary to highlight the knowledge gaps in the existing literature before introducing your work in the last paragraph of the introduction.

3.      As seen in your previous paper the TA1 titanium material presents an anisotropy. The TA1 titanium sheet is anisotropic material due to the manufacturing process (rolling direction , transverse direction) and this anisotropy should be taken into account in the simulation. How authors account this anisotropic behavior in the numerical model.

4.      The mechanical properties of material used in the simulation should be added and should be identified with a formula description

5.      Software used in Numerical simulation should be added in the manuscript

6.      More details for the FEA should be added i.e (type of element used; number of elements).

7.      A comparison between actual geometry and the desired one of the deformed specimens should be added for different wall angle.

8.      The accuracy of the numerical model should be investigated by presenting a comparison between numerical and experimental results.

9.      What is the main idea of the perforated plate, a comparison between perforated and non-perforated plates should be added in the manuscript.

10.  Recently published article should be added to improve the literature review. It is recommended to add the references of significant articles recently published developing SPIF process and plasticity

*The incremental sheet forming; technology, modeling and formability: A brief review, Proceedings of the Institution of Mechanical Engineers Part E Journal of Process Mechanical Engineering, 2022, 10.1177/09544089221093306.

*SPIF Manufacture of a Dome Part Made of AA1060-H14 Aluminum Alloy Using CNC Lathe Machine: Numerical and Experimental Investigations, Arabian Journal for Science and Engineering, 2021, https://doi.org/10.1007/s13369-021-05919

*Numerical Formulation of Anisotropic Elastoplastic Behavior Coupled with Damage Model in Forming Processes, Mathematics 2023, 11, 204. https://doi.org/10.3390/math11010204

*Accuracy of Variational Formulation to Model the Thermomechanical Problem and to Predict Failure in Metallic Materials, Mathematics, 2022, 10(19), 3555;

According to my opinion, the mentioned major issues must be corrected in the paper to have a scientific impact.

Author Response

Dear editors and reviewers,

Thank you for your letter and for the reviewers’ comments concerning our manuscript entitled “Research on the Formability of TA1 Titanium Mesh in Single Point Incremental Forming” (Materials-2266352).

Those comments are all valuable and very helpful for revising and improving our paper, as well as the important guiding significance to our researches. We have studied comments carefully and have made correction which we hope meet with approval. Revised portion are marked in red in the paper. The revision was addressed point by point below. 

Reviewer 2 Report

Dear Authors,

The Materials journal is a reputable, high-standard journal. Your manuscript requires a lot of work and time to meet the requirements of publication in a such reputable journal.

See the comments below.

Introduction

Check the formatting of references to literature. They shouldn't be superscripted.

II. Titanium Mesh Incremental Forming Experiment Analysis

All descriptions in the table 1 should start with a capital letter.

“In the experiment, square titanium mesh (120×120×1mm) were used.”

Comment: Please provide chemical composition and mechanical properties of the titanium grade.

“The toolpath was designed using the CAM software,…”

Comment: What exact software was used?

“…the anti-wear hydraulic oil jet was …”

Comment: Provide more details: oil designation, manufacturer, etc.

Figure. 3

Comment: Dot is not needed.

“Figure 4. The Titanium Mesh Formed by Constant wall angle Model.”

Comment: Since there are 6 images in the figure, the caption is not complete. Complete the caption to indicate the differences in individual images.

“2.3. Microhardness Test

For the purpose of further exploring the effect of SPIF on the material hardness…”

Comment: So did you measure hardness or microhardness? What was the load? What was the dwell time?

“Figure 5. The microhardness test result for titanium mesh incremental forming circular truncated  Cone workpiece.”

Comment: Since you do not provide the full and proper hardness or microhardness designation in the Fig. (e.g. 100 HV0.05), it should be indicated at least in the caption.

“the hardness of the inner surface by 26.59 percent and the hardness of the outer surface by 16.87 percent.”

Comment: It does not make sense to give percentages of change in hardness to the nearest hundredth. Are the values a mean value? How many measurements have you conducted?

“Figure 6. The SEM fracture scan imagery for titanium mesh sample with wall angle of 60°.”

Comment: The caption is not complete (there are 6 images). See the comment above. What is the different between Fig. 6b to 6e ? The last image is not described at all.

“Figure 8. Typical record of the three components of the process force versus (Forming depth 10 mm, 314 thickness 1 mm, tool diameter 10 mm, wall angle 45°, step-down size 1 mm).”

Comment: Versus what?

“Figure 9 represent the evolution…”

Comment: Check grammar.

“… the forming process of titanium mesh was to form the truncated cone at the wall angles of 25°, 30°, 35°, 40°, and 45°..”

“Figure 10. Comparison Diagram Regarding the Maximum Principal Stress and Equivalent Stress of 412 Different Wall Angles.”

Comment: Check editing.

“Figure 12. Comparison Diagram Regarding the Principal Strain and Effective strain Along X Axis of Different Wall Angles.”

Comment: There are 6 graphs in the figure that are not even labeled (a-n-d). Label them and describe them in detail.

“Figure 13. The Titanium Mesh Circular Truncated Cone Workpiece Contour of Different wall an- 474 gles.”

Comment: Why do you start words with a capital letter? The caption does not correspond at all to what is shown in the picture.

“Figure 14. The maximum offset for X-Direction and Y-Direction of forming titanium mesh.”

Comment: The caption is confusing and incomplete. There are 3 graphs referring to X,Y and Z!

“Figure 16. Changing curve regarding wall thickness and thinning ratio of the titanium mesh along 524 X Axis.”

Comment: There are 2 images on it. The caption is confusing.

“Figure 18. Extreme value changing curve regarding the titanium mesh wall thickness and thinning 547 ration under the conditions of different wall angles.”

“Figure 19. The trend of Horizontal force and Vertical force with the varying wall angle of titanium 581 mesh.”

Comment: See above comment. All captions are confusing.

 References

Comment: There is only one item published in 2022! Other items are out of date.

Author Response

(The authors gave the same response as above.)

Reviewer 3 Report

REVIEW

on article

Research on the Formability of TA1 Titanium Mesh in Single Point Incremental Forming

Li Ruxiong, Wang Tao and Li Feng

SUMMARY

The article submitted for review is devoted to a topical issue. It presents studies of the formability of the TA1 titanium mesh in single point incremental forming (SPIF). The authors focused on the fact that in the analysis's light of the titanium mesh single point overmolding principle of SPIF and the corresponding features in the forming process; it can be found that the wall angle of the cone of the cross section is a key parameter affecting the quality of the overmolding. It is also a key evaluation indicator for studying the application of incremental forming technology on a complex surface. Thus, the authors have chosen methods of experiment and modeling by the finite element method. The authors obtained several important results, and their study is interesting and useful in terms of scientific novelty and practical significance.

At the same time, the article has several shortcomings. They need to be corrected. The reviewer's comments are presented below.

COMMENTS

1) The first observation is that the authors did not formulate the research problem. They only reported that the wall angle of the cross-sectional cone is a key parameter affecting the quality of overmolding, and all this leads to the fact that the assessment depends on it for studying the application of incremental forming technology on a complex surface. However, this is the relevance of the study. The problem of research should be what scientific deficit currently exists in this industry. Authors need to add the scientific problem statement to the beginning of the abstract.

2) In addition, the authors talk about the fact that there is an effective deformation of the shaping force, the friction energy increases as the wall angle increases, the growth rate of the horizontal shaping force is much higher than the axial shaping force. At the same time, quantitative characteristics of the achieved result are not given. The abstract should reflect not only a qualitative picture of the study, but also a quantitative picture. Authors need to add information in numerical terms. Thus, the abstract needs to be improved.

3) Further, the authors in the keywords and in the title of the article give the abbreviation TA1. Probably, the authors should immediately give a decoding of this term in order to interest the number of readers who do not know this wording. This will promote interest in the article and increase the rating of the author and the journal as a whole. Authors are advised to avoid a large number of abbreviations that are not deciphered in the text.

4) The literature review presented in the "Introduction" section is not well done. Authors need to analyze more sources to clearly reflect scientific novelty. The authors analyzed 21 references in the introduction, although there is a much larger number of studies on the topic of titanium meshes and their formation. Therefore, the authors are recommended to bring the number of analyzed references to at least 30, so that the introduction reflects the scientific novelty. In addition, it is necessary to clearly formulate the purpose and tasks of the study at the end of the "Introduction" section.

5) The authors used various research methods, both experimental and modeling. However, they did not present a research program. The authors need to eliminate this deficiency and add a flowchart with a program of experimental studies and modeling studies to the beginning of section 2.

6) The SEM analysis in Figure 6 seems interesting, but it is poorly explained. More justification for the results seen should be added.

7) In figure 7 there are poorly readable or indistinguishable characters. All graphic images should be checked again for their quality.

8) I would like to see a more detailed rationale for Figures 10 and 11. Probably, Figure 12 should be given in a better quality.

9) In general, section 3 looks like a kind of scientific protocol. The authors cited a lot of graphic material, but there was little discussion between them, as a result of which the style of presentation suffers. The authors need to make some stylistic changes in section 3 in particular and in the article as a whole.

10) The discussion of the obtained results is poorly done. Authors need to provide a detailed explanation of the effect obtained and compare their results with those of other authors.

11) Conclusions should be specified in terms of scientific novelty and scientific result. It should also reflect the prospects for the development of the study and the prospects for the practical application of the results.

12) As mentioned above, the authors analyzed only 23 references on the topic of the study, but this is very small, due to the fact that such studies on titanium meshes are carried out all over the world. The authors need to significantly work on the list of references and its analysis. The list of references should be increased to 35-40 references throughout the article.

13) Thus, the general conclusion of the reviewer is as follows. The article is interesting, useful and relevant. Those comments made by the reviewer are not critical, but the authors are strongly encouraged to correct them. After their correction, the article can be published in the "Materials".

Author Response

(The authors gave the same response as above.)

Round 2

Reviewer 1 Report

Point 1: the authors don’t explain correctly the improvement compared to their previous work published in (Metals. 2022, 12, 1944). Also, what is the difference between perforated TA1 Titanium and TA1 Titanium mesh.

Point 2: references, illustrating dental implant, added are not adequate and bad presented in the introduction (refs 1, 2, 3).

Point 3: Rewrite sentence:

Line 15: Most of the previous studies study focuses on conventional plates.

Line 18: the analysis and reach on SPIF for titanium mesh objects

Line 66: its enhanced formability, flexible manufacturabilityand reduced forming force

Point 4: Lines 35 to 39 are not necessary in abstract section and should be presented in results

Point 5: Remove this sentence: The study presented in this paper began in 2017.

Point6: Figure illustrating the predicted tensile curves for different direction (0, 45, 90) and experimental tensile curves should be added to comfirm the accuracy of Hill parameter’s used in the simulation.

Point 7: Two figures named Fig 3

Point 8: in this paper we found the same results as presented in reference  ( Li, R.X.; Wang, T. Research on Single Point Incremental Forming Characteristics of Perforated TA1 Sheet[J]. Metals. 2022, 12, 1944), see figures 1, 2, 3, 3bis, 12

Author Response

Thank you for your useful comments and suggestions on our manuscript. We have revised carefully the paper and addressed all these comments. Detailed corrections have been listed below point by point. In addition, we highlighted the changes with red colored text.

Reviewer 3 Report

All my comments were considered, and corrections were done.

I recommend the article for publishing.

Author Response

Thank you very much for your positive evaluations and the valuable suggestions on our manuscript. We tried our best to improve the manuscript and made some changes in the manuscript. These changes will not influence the content and framework of the paper. We appreciate for Editors/Reviewers’ warm work earnestly, and hope that the correction will meet with approval. Once again, thank you very much for your comments and suggestions.
